# A diminutive perinate European Enantiornithes reveals an asynchronous ossification pattern in early birds

Fabien Knoll [1,2], Luis M. Chiappe[3], Sophie Sanchez [4,5], Russell J. Garwood [2,6], Nicholas P. Edwards[2,7], Roy A. Wogelius[2], William I. Sellers [2], Phillip L. Manning[2,8], Francisco Ortega [9], Francisco J. Serrano [3,10], Jesús Marugán-Lobón[3,11], Elena Cuesta [11], Fernando Escaso [9] & Jose Luis Sanz [11]

Fossils of juvenile Mesozoic birds provide insight into the early evolution of avian development, however such fossils are rare. The analysis of the ossification sequence in these early-branching birds has the potential to address important questions about their comparative developmental biology and to help understand their morphological evolution and ecological differentiation. Here we report on an early juvenile enantiornithine specimen from the Early Cretaceous of Europe, which sheds new light on the osteogenesis in this most species-rich clade of Mesozoic birds. Consisting of a nearly complete skeleton, it is amongst the smallest known Mesozoic avian fossils representing post-hatching stages of development. Comparisons between this new specimen and other known early juvenile enantiornithines support a clade-wide asynchronous pattern of osteogenesis in the sternum and the vertebral column, and strongly indicate that the hatchlings of these phylogenetically basal birds varied greatly in size and tempo of skeletal maturation.

[1] ARAID—Fundación Conjunto Paleontológico de Teruel-Dinopolis, 44002 Teruel, Spain. [2] School of Earth and Environmental Sciences, University of Manchester, Manchester M13 9PL, UK. [3] The Dinosaur Institute, Natural History Museum of Los Angeles County, Los Angeles, CA 90007, USA. [4] Department of Organismal Biology, Uppsala University, 752 36 Uppsala, Sweden. [5] European Synchrotron Radiation Facility, 38000 Grenoble, France. [6] Department of Earth Sciences, Natural History Museum, London SW7 5BD, UK. [7] Stanford Synchrotron Radiation Lightsource, SLAC National Accelerator Laboratory, Menlo Park, CA 94025, USA. [8] Department of Geology and Environmental Geosciences, College of Charleston, SC 29424 Charleston, USA. [9] Facultad de Ciencias, Universidad Nacional de Educación a Distancia, 28040 Madrid, Spain. [10] Facultad de Ciencias, Universidad de Málaga, 29010 Málaga, Spain. [11] Facultad de Ciencias, Universidad Autónoma de Madrid, 28049 Madrid, Spain. Correspondence and requests for materials should be addressed to F.K. (email: knoll@fundaciondinopolis.org)

An organism's ontogeny informs the study of its evolutionary history. Developmental sequences—a prime example being the sequence of ossification in vertebrates—constitute a rich data source for trait evolution[1,2]. The timing of ossification can potentially address many important questions in the field of comparative biology. As such, temporal analysis of cartilaginous and skeletal development has recently attracted significant attention[3]. Birds are model organisms for surveying ossification sequences: not only have they large, easily accessible eggs, but they also show a high degree of bone fusion to strengthen the skeleton, which must withstand the stress generated by flight[4]. Although bone ossification sequences are well known in a number of extant bird species, there is a dearth of data regarding skeletogenesis in basal avians, which results from the extreme rarity of fossil avian embryos or hatchlings. This considerably limits the palaeontological insight into, and therefore the reliability of, different potential models of developmental character evolution.

Here we describe an early juvenile Enantiornithes, MPCM-LH-26189 a/b, from the Early Cretaceous Las Hoyas deposits of Spain. The specimen is important because it died around the time of birth: a critical stage for the examination of osteogenesis in birds. In common with modern birds, ossification was not completed in enantiornithine hatchlings[5]. Therefore, this perinate provides valuable insight into the spatial patterning of osteogenesis in Enantiornithes, the most speciose Mesozoic avian clade and the only non-ornithuromorphs with extensive skeletal fusion[6]. Comparing the osteogenesis in this fossil with the other few early juvenile enantiornithine specimens published to date supports an asynchrony between the osteogenesis of sternum and that of the vertebral series. This asynchronous pattern implies a great variation in skeletal maturation in the hatchlings of these basal birds.

## Results

**Morphological description**. The new fossil we report here, MPCM-LH-26189 a/b (slab and counterslab housed at the Museo

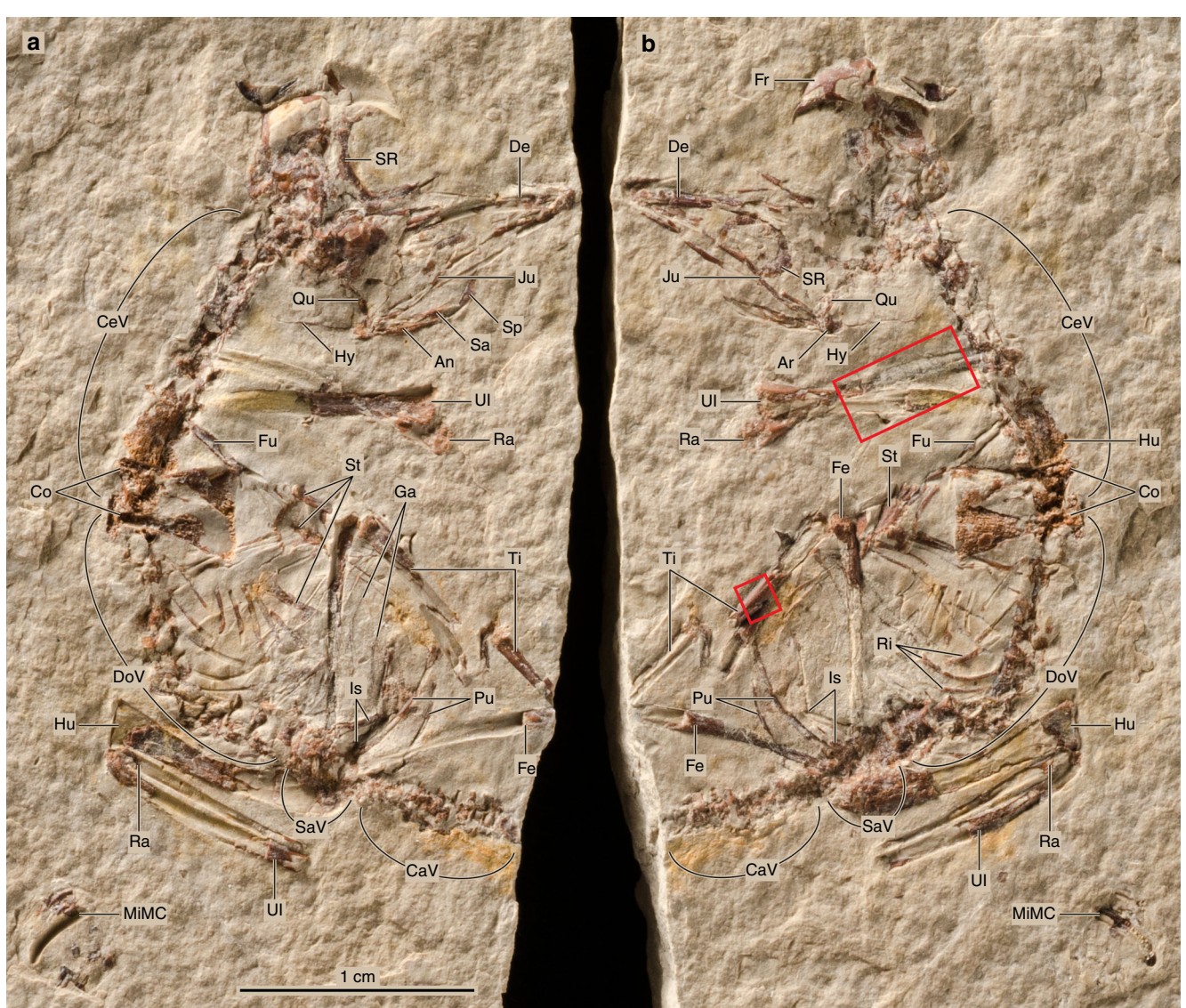

**Fig. 1** Overview photographs of the slab and counterslab of MPCM-LH-26189. Slab **a** is on the left, slab **b**, on the right. The two red boxes indicate the localisation of the areas analysed histologically (Fig. 3). Abbreviations: An: angular, Ar: articular, CaV: caudal vertebrae, CeV: cervical vertebrae, Co: coracoid, De: dentary, DoV: dorsal vertebrae, Fe: femur, Fr: frontal, Ga: gastralium, Hu: humerus, Hy: hyoid, Is: ischium, Ju: jugal, MiMC: minor metacarpal, Pu: pubis, Qu: quadrate, Ra: radius, Ri: rib, Sa: surangular, Sp: splenial, SaV: sacral vertebrae, SR: sclerotic ring, St: sternum, Ti: tibia, Ul: ulna

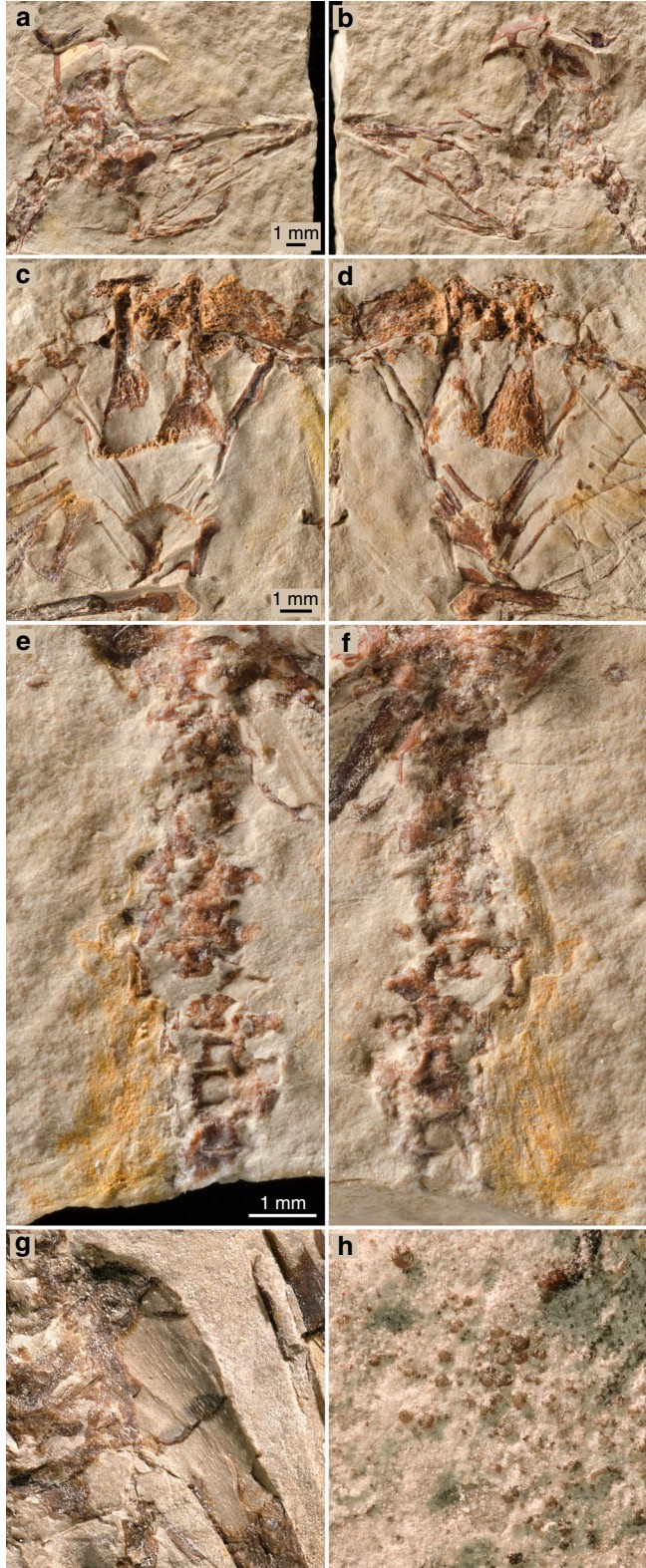

**Fig. 2** Details of MPCM-LH-26189. **a** Skull on slab a. **b** Skull on slab b. **c** Sternum on slab a. **d** Sternum on slab b. **e** Tail on slab a. **f** Tail on slab b. **g** Impression into the matrix of the pattern of longitudinal striation of the bone surface of the proximal humerus. **h** 'Corpuscles' in the pelvic area

de Paleontología de Castilla-La Mancha, Cuenca, Spain), consists of a nearly complete and largely articulated skeleton (Fig. 1). The feet, most of its hands, and the tip of the tail are the only missing parts.

The skull (Fig. 2a, b) is large in proportion to the body. It is partially crushed, and its bones largely disarticulated. Synchrotron microtomography shows an absence of bones buried in the matrix of the slabs. The braincase has been fractured along the interfrontal and interparietal sutures, which was facilitated by the fact that cranial elements such as the frontals and parietals are not fused with one another in most enantiornithines[7]. Its left half is nicely preserved on one slab (Fig. 1b) and in its approximate original location, whereas the right half, strongly displaced caudally, is best viewed on the other slab (Fig. 1a). The frontals and the parietals form a uniformly curved cranial vault, which is separated from the supraoccipital plane by a low nuchal crest. The ventral surface of the left frontal (orbital roof) is only gently arched, suggesting large orbits. Large scleral rings are preserved close to their original position inside the orbit. Parts of both frontals (which are very thin, suggesting little or no pneumatisation) have fallen out, uncovering a cerebrocast with very slight inflation. In this regard, the cerebral anatomy of MPCM-LH-26189 falls in between that of the basal avian *Archaeopteryx*[8], in which the telencephalon is even less swollen, and the putative basal ornithurine *Cerebavis*[9], whose telencephalic expansion is close to that seen in most extant birds (*Archaeopteryx* and *Cerebavis* are the only two Mesozoic birds for which reliable reconstructions of the endocranial casts are available). A delicate slightly curved bone situated close to the left dentary is interpreted as the jugal bar. The straight and uniformly wide dentary exhibits a limited number of nutrient foramina close to its dentigerous margin. Postdentary bones (including splenial, surangular, angular, and articular) are also preserved, although not in their original relative position. One element of the hyoid apparatus, the ceratobranchial, is visible caudal to the jaw.

The cervical series, exposed dorsally, appears to be composed of 9 vertebrae (the cervical-dorsal transition is overlapped by the coracoids), a number that falls within the range shown by other Enantiornithes[10]. The postzygapophyses and prezygapophyses of these vertebrae project at a significant angle with respect to the sagittal plane, forming an X-pattern in dorsal view (see fig. 10 in ref. [5]). At least 10 thoracic vertebrae are laterally exposed and preserved in articulation. As in other Enantiornithes, the cranial and caudal articular surfaces of the thoracic vertebrae are nearly flat. The thoracic centra appear laterally excavated by a broad but relatively shallow fossa, a condition common to other Enantiornithes[10]. The prezygapophyses of the mid-thoracic vertebrae extend beyond the cranial articular surface and their neural spines are broad and subrectangular. Thoracic ribs are articulated to the thoracic vertebrae; they lack uncinate processes (neither free nor fused) and gradually expand towards the articulation with the sternal segment of the rib. There appears to be six or seven pairs of sternal ribs, although it is not clear whether all of them articulated with the sternum (the articular facets would have been contained within a portion of the sternum that was still cartilaginous). The sacrum appears to be composed of 5–6 vertebrae; fewer than the 8 typical of the fully fused synsacrum of adult Enantiornithes, but similar to the 6 reported for other juveniles of this clade[5]. The cranial synsacral vertebae, exposed ventrally, are unfused to one another. The tail is formed by at least 10 individualized vertebrae (Fig. 2e, f); because the end of the tail is missing, nothing can be said about the presence or absence of a pygostyle. The preserved caudal vertebrae have flat articular ends and some of them have triangular to rhomboid haemal arches.

The two coracoids, the furcula, and three sternal ossifications (Fig. 2c, d) are preserved and exposed ventrally in one slab and dorsally in the other. No scapula is present. The coracoids are strut-like; their lateral and medial margins are relatively straight. The dorsal and ventral surfaces of the coracoids are concave and

convex, respectively. A dorsal coracoidal fossa is present in many Enantiornithes[10]. The sternal margin of these bones is straight. The furcula is Y-shaped, as in other Enantiornithes[10]; its hypocleidum has an oval section and a preserved length that is less than half the length of the rami. As in other Enantiornithes, the ventral margin of the furcular rami is much broader than the dorsal margin[10]; the lateral surface forms a trough that runs throughout the length of the rami. The sternum is represented by three ossifications: two elongate and caudally "booted" lateral trabeculae and a fan-shaped median element. The distal ends of the lateral ossifications exhibit a prominent expansion that

resembles the triangular distal expansion of the lateral trabeculae (lateral process) of many adult enantiornithine birds[10]. On the cranial end of the left lateral ossification, there is a laterocranially projecting process. It is unclear whether this process would be incorporated into the body of the ossified sternum in later ontogenetic stages or whether it would remain as a lateral projection in the adult sternum. The ventral surface of the median element is strongly convex, indicating the existence of an incipient carina (i.e. sternal keel). The position of the sternal ossifications appears to be close to their natural relation to one another. On the basis of this interpretation, the lateral

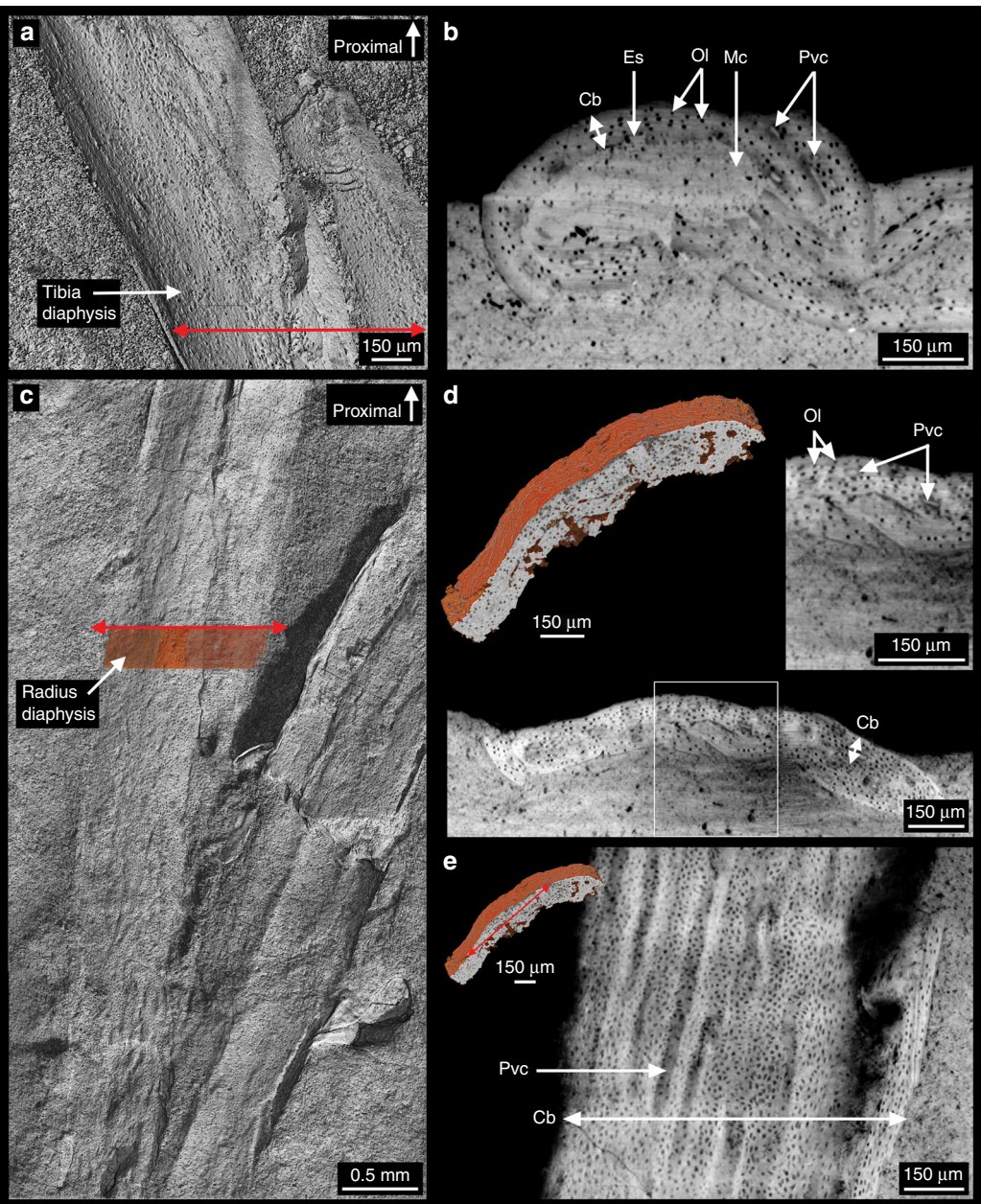

**Fig. 3** Virtual long bone histology of MPCM-LH-26189 b. **a** 3D model of the tibia diaphysis (the red double arrow indicates the location where the virtual thin section in b was made). **b** Transverse section of the tibia showing a thin, poorly vascularised cortex and cell lacunae in black in the cortex. The inner region of the cortex is eroded, thereby giving place to a large medullary cavity, which has been infilled in with rock matrix. This matrix has multiple inclusions of different densities, but is generally similar in grey value to the bone, due to a similar density and thus X-ray attenuation coefficient. **c** 3D model of the radius diaphysis (the ochre band with a red double arrow on top indicates the location of the sectioned portion shown in (**e**)). **d** Transverse section of the radius showing a microstructure similar to that of the tibia, although with a slightly more vascularised cortex (the cortex and medullary cavity are crushed). **e** Longitudinal section of the radius (made according to the red double arrow in the inset) showing the longitudinal orientation of the vascularisation. Abbreviations: Cb: cortical bone, Es: erosion surface, Mc: medullary cavity, Ol: osteocyte lacunae, Pvc: primary vascular canals

ossifications would have projected farther caudally than the median xiphoid process (e.g. *Protopteryx*[11]).

Both humeri, ulnae, and radii are preserved, although the left arm and forearm are displaced and parallel to the sacrum. The distal end of the left humerus shows a dome-shaped ventral condyle and a flatter dorsal condyle. Both ulna and radius are relatively straight. The ulnar shaft is approximately twice as wide as the radius. A fragmentary set of metacarpals is preserved close to the left humerus, radius, and ulna, suggesting that it belongs to this side. The best preserved metacarpal is gently curved distally. It strongly recalls the minor metacarpal (III) of other Enantiornithes, and is, therefore, identified as such.

The morphology of the pelvis is difficult to interpret, as the bones are not in their natural position. Two long, slender, and gently curved elements, more or less parallel to each other, are recognized as the pubes. These bones are laterally compressed and one of them (small slab) exhibits a distal expansion. Distally expanded pubes are common among Enantiornithes[10]. The other pair of bones, here interpreted as the ischia, are broader and more belt-like. These bones are somewhat shorter than the pubes. The two femora and portions of both tibiae are preserved, but no fibula is visible. The femora are long and slender; the cranial distal surface lacking a patellar groove, as in other enantiornithines. Remnants of a typical zig-zag gastral basket, with a pair of overlapping gastralia on each side, are preserved in front of the pelvic bones and aligned with the sternum. Enigmatic numerous diminutive "corpuscles" in the pelvic area (Fig. 2h) were probably originally held in the digestive tract. MPCM-LH-26189 can be referred to the Enantiornithes on basis of the lateral excavation of the furcular rami[10]. Additionally, MPCM-LH-26189 shows a pattern of sternal osteogenesis identical to well-preserved enantiornithine skeletons from the Early Cretaceous of China[5,12,13].

**Osteohistological analysis**. MPCM-LH-26189 is extremely small (length of the cervico-sacral vertebral series: ∼29 mm). It is indeed clearly smaller than any other complete or nearly complete juvenile enantiornithine specimen published to date, although its wings are larger than the most complete isolated wing recently described from mid-Cretaceous Burmese amber (the ulna is 24.7% larger)[14]. The impression of the proximal left humerus in the matrix shows a dense pattern of longitudinal grooves (Fig. 2g). Such grooves have been shown to correspond to primary cavities (Fig. 3e), which open onto the surface of the cortex in young and fast-growing bone[15,16]. These are consistent with a perinate ontogenetic stage[5,17,18], as supported by the general proportions of MPCM-LH-26189 (i.e. very-large head compared to body size and relatively large orbits).

Virtual cross-sections in the shaft of the tibia and radius show very-thin cortices (Fig. 3a–d). The presence of an irregular surface in the inner region of the cortical bone of the tibia, which is well preserved in three dimensions, suggests internal erosion (Fig. 3b). This explains the thinness of the primary bone and the large extension of the free medullary cavity (Fig. 3b). There is no evidence of secondary deposition on the inner surface of the cortical bone. The cortical bone is pierced by primary vascular canals and exhibits a large number of globular-shaped osteocyte lacunae (Fig. 3b, Supplementary Fig. 1). These cell lacunae are relatively large (7–13 μm). No lines of arrested growth can be observed. Although flattened, the radius displays a similar microstructural organisation (Fig. 3d, e) with an obvious longitudinal orientation of the vascularisation (Fig. 3e). Previous examinations have shown that the osteohistology of Enantiornithes is characterized by the presence of a thin and relatively compact cortical wall that comprises large cell lacunae and

surrounds an extended free medullary cavity[19–26]. Lines of arrested growth have been observed in subadult and adult Enantiornithes[19–23,26–28], but none is present in the cortex of the tibia and the radius of MPCM-LH-26189 (Fig. 3b, d), indicating that the bony tissue was deposited during a relatively fast-growing period (i.e. as a "zone", before the formation of a line of arrested growth). This, together with the primary nature of the vascularisation, the round shape of the osteocytes lacunae and the uneven peripheral margin of the medullary cavity (with no endosteal bone), strongly suggests that the bone was actively growing, characteristic of an early stage of development, when the bird died. The resorption front bears testimony to an ongoing intensive restructuring of the bone, progressively integrating the vascular canals into the medullary cavity (Fig. 3b). The vascular density in MPCM-LH-26189 is much lower than that of the embryonic tibia of an enantiornithine embryo from the Late Cretaceous of Mongolia[21,29], but higher than that of the adult (or subadult) individual referred to as cf. *Concornis lacustris*[27] or other enantiornithine specimens[19–24,26,30].

Long bone histology of MPCM-LH-26189, therefore, strongly suggests that the appendicular skeleton is at an early stage of its development. The compactness of the bone (Fig. 3b, d), however, rules out the hypothesis of any remnants of cancellous embryonic tissue in the long bones of this specimen. Histological investigations on enantiornithines have been focused on specimens that generally represent widely separated ontogenetic stages (embryo or adult). MPCM-LH-26189 is important in that it provides histological information corresponding to an intermediate developmental stage, certainly an early posthatchling juvenile stage. Based on its humeral length (which is a better proxy for mass estimates in birds than the femoral length), MPCM-LH-26189 would have weighed ∼10.3 g according to one formula (see Table 2 in ref. [31]) and ∼9.8 g according to another (see Table 2 in ref. [32]).

**Discussion**

Enantiornithes represent one of the earliest avian evolutionary divergences, in which the sternum adopts an elaborate morphology, strikingly different from the simple paired plate-like sternum of other non-ornithothoracine birds, such as *Confuciusornis*, *Jeholornis*, and a variety of non-avian dinosaurs[12]. In extant birds, the sternum is one of the last elements to ossify, and it does so from several ossification centres[33]. Zheng et al.[12] recognized that the ossified sternum of most adult Enantiornithes (except for the most basal forms[13]) formed from the fusion of four to six elements, most typically including a fan-shaped caudomedial ossification (xiphial region), two parasagittal strap-like trabeculae, and a craniomedial bony discoid. The ossification pattern seen in the sternum of MPCM-LH-26189 (Figs. 2c, d and 4b) is complex and characteristic of early juvenile enantiornithines: both the caudomedial ossification and the trabeculae are present, although the craniomedial disc had yet to ossify (see Fig. 3 in ref. [12] and Fig. 4 in ref. [13]). In a few adult enantiornithines, such as the holotypes of *Concornis lacustris* and *Rapaxavis pani* (see fig. 8 in ref. [34]), but not MPCM-LH-26189, two additional craniolateral subtriangular elements (i.e. paracoracoidal ossifications) are found.

In stem-group birds, like in extant taxa, there are a higher number of free caudal vertebrae in the juvenile stages of any given species[35–37]. In adult Enantiornithes, no more than eight free caudal vertebrae precede the pygostyle[10]. The last caudal vertebra may appear partially incorporated into the pygostyle[38]. However, MPCM-LH-26189 shows at least ten free caudal vertebrae (Fig. 2e, f) and the fracture of the slabs across the tail makes it possible that even more free vertebrae were originally present

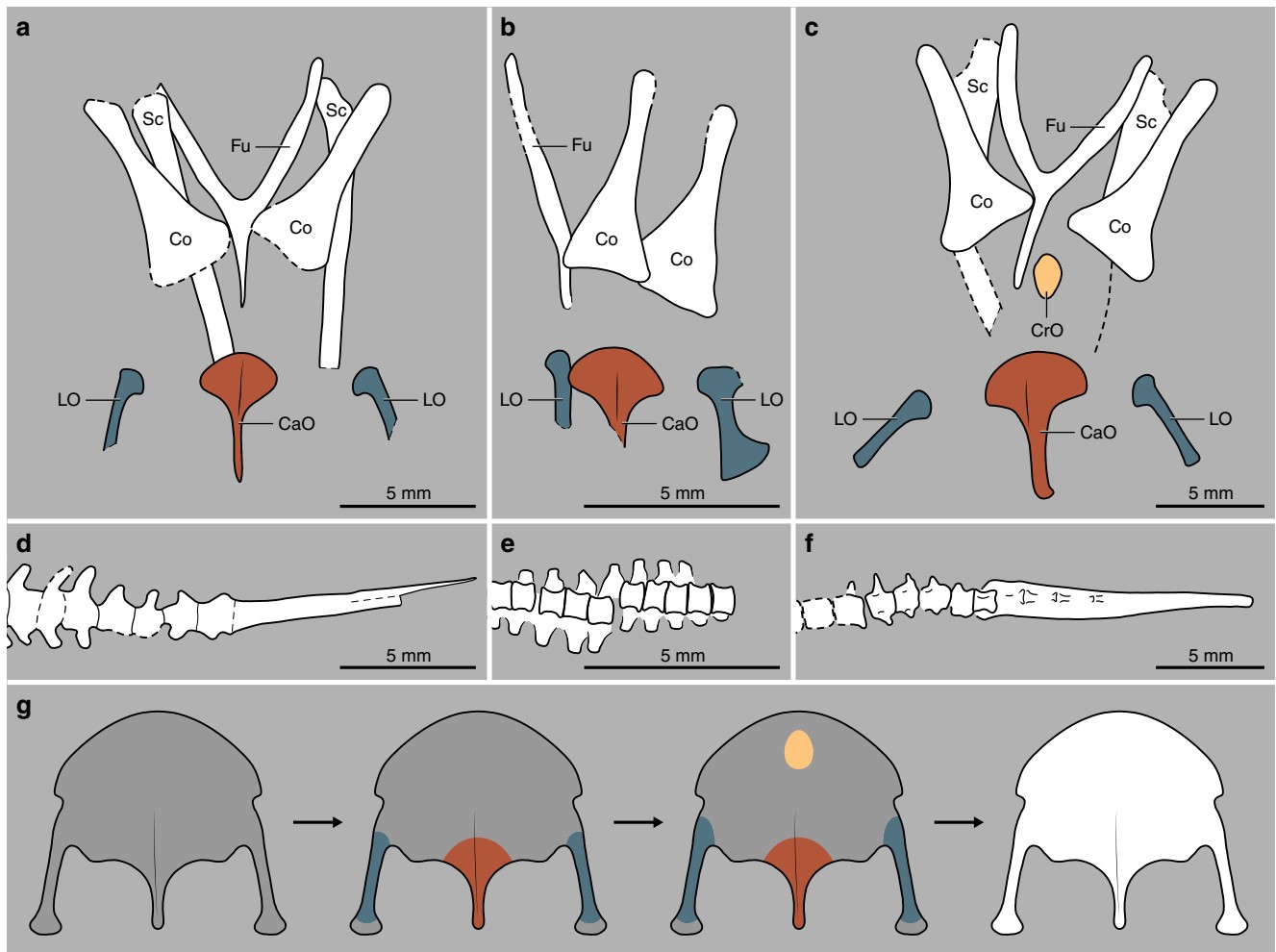

**Fig. 4** Interpretative drawings showing the asynchrony between the development of sternum and tail in Enantiornithes. **a** Thoracic girdle of GMV-2156/NIGP-130723; **b** thoracic girdle of MPCM-LH-26189; **c** thoracic girdle of GMV-2159; **d** caudal series of GMV-2156/NIGP-130723; **e** caudal segment of MPCM-LH-26189; **f** caudal series of GMV-2159; **g** sternogenesis in derived Enantiornithes: dark grey indicates cartilage and other colours, bone (after ref. [12]). Abbreviations: Fu: furcula, CaO: caudal ossification, Co: coracoid, CrO: cranial ossification, LO: lateral ossification, Sc: scapula

more caudally (Fig. 2e, f). We suggest that this unusually high number of free caudal vertebrae is due to the immature stage of the specimen (Fig. 5); one or several cranial-most and/or caudal-most vertebrae would have been fused with the synsacrum and/or pygostyle as the skeleton of MPCM-LH-26189 matured into adulthood.

MPCM-LH-26189 shows that skeletal ossification was still incomplete in the early postnatal phases of Enantiornithes. Differences in the ossification of the sternum and the number of free caudal vertebrae in MPCM-LH-26189, when compared to other juvenile enantiornithines, reveal a clade-wide asynchrony in the sequence of ossification of the sternum and tail (Table 1 and Fig. 4). For example, while MPCM-LH-26189 is only two-thirds the overall size of GMV-2156/NIGP-130723 (see ref. [5]), it has a similar degree of sternal ossification (i.e. lateral trabeculae and caudomedial ossification) but a lesser degree of tail ossification (i.e. at least three more free caudal vertebrae) than the latter specimen. GMV-2159 (see ref. [5]) and STM 34-1 (see ref. [12]), specimens that are about twice the size of MPCM-LH-26189, reveal a similar degree of tail ossification as GMV-2156/NIGP-130723 but a greater ossification of the sternum (i.e. presence of a craniomedial disc; Fig. 4). The recently described UFRJ-DG 031 Av[39] is about one-third larger than MPCM-LH-26189 and smaller than GMV-2156/NIGP-130723, yet it has eight free

caudal vertebrae (similar to the seven or eight of GMV-2159) and its sternum was presumably completely cartilaginous. In a pellet from Las Hoyas containing four early juvenile enantiornithines, one of the two largest individuals has seven free caudal vertebrae followed by a large pygostyle, while the other has a minimum of 12 free caudal vertebrae lacking a fully formed pygostyle[40]. All these observations are consistent with independence between the ossification sequences of the sternum and the vertebral column in early juveniles Enantiornithes. In fact, in having a timing of ossification of the sternum that was highly variable between species and independent of the remaining bones of the skeleton, Enantiornithes show an osteogenetic pattern that is as variable as that of extant birds[33,41–44].

Information regarding early developmental stages of stem fossil birds provides important data as hatchling maturity is related to life history traits such as the broad spectrum of developmental modes (precocial-altricial) that characterize the diversity of living birds.[33] Based on a wealth of evidence, including the presence of fledged wings, large brains, advanced degree of ossification, forelimb proportions, and long bone histology of perinates, previous authors have consistently argued that enantiornithines hatchlings were highly precocial[5,14,21,45–48]. While the skeleton of extant precocial hatchlings generally exhibits a higher degree of ossification than that of altricial hatchlings[41,42,49], data on

neonate ossification are still deficient and inconclusive for predicting specific developmental modes[50,51]. This notwithstanding, it is clear that locomotion requires a high degree of ossification to strengthen the skeleton against mechanical stresses, which provides a reason for the greater degree of skeletal ossification in the more mobile precocial hatchlings. Such correlation between ossification and mechanical demands may be particularly true for the sternum, which would provide a weaker anchor to the wing muscles if partially ossified[52]. Therefore, the largely cartilaginous sternum of MPCM-LH-26189 hints at functional limitations in

terms of flying ability, which should not be taken as evidence of altriciality given that semiprecocial and many precocial species are able to walk at an early age, but are unable to fly until almost fully grown[53].

The evolutionary diversification of birds has resulted in a marked variation in locomotory adaptations of perinates, a wide range of hatchling developmental strategies, and important differences in their growth rates. The disparate asynchronic patterns of ossification of perinate enantiornithines suggest that the developmental strategies of these basal birds may have been more diverse than previously thought. Because developmental modes of birds are tied to ecological conditions, our findings are in line with recent results suggesting greater ecological diversity and varied locomotor behaviours in Enantiornithes[38,54].

## Methods

**Taxonomical identification.** The Early Cretaceous wetland deposits (La Huérguina Formation, c. 127Ma) of the Las Hoyas fossil site[55,56] have yielded a diversity of enantiornithine taxa including a range of post-hatching developmental stages[27,57,58].

Three species of Enantiornithes have been recognized so far from the Las Hoyas fossil site[59]. They are based on one single specimen each and none of them preserves any cranial element. MPCM-LH-26189 does not pertain to *Eoalulavis hoyasi*[60], in which the sternum is oblanceolate. However, whether it belongs to *Concornis lacustris*[61], which is much larger but an adult, *Iberomesornis romerali*[62], which is not so much larger, but is also a juvenile, or instead it represents a new species is difficult to ascertain. The cranial margin of the median element of the sternum is uniformly convex in MPCM-LH-26189, which might suggest that no paracoracoidal ossification would eventually develop, in contrast with the situation in *Concornis*. The humerus/ulna length and humerus/femur length ratios are 8.1% greater and 5.8% lesser, respectively, in MPCM-LH-26189 than in *Iberomesornis*, whereas they are 9.2% greater and 24.0% lesser, respectively, in MPCM-LH-26189 than in *Concornis* (Supplementary Table 1). MPCM-LH-26189 is, therefore, closer to *Iberomesornis* than to *Concornis* in long bone proportions, although we recognize that this does not constitute a reliable argument of taxonomic closeness given the different ontogenetic stages of the three specimens compared.

**Histological analysis.** We have undertaken virtual histological investigations by means of propagation phase-contrast synchrotron microtomography[63]. The experiment was performed at the beamline ID19 of the European Synchrotron Radiation Facility. The specimen was imaged at submicron resolution (with a voxel size of 0.719 μm) to observe the bone microstructures. Virtual thin sections were made in the shaft of the left tibia and right radius of MPCM-LH-26189 using VGStudio MAX 2.2 (Volume Graphics, Heidelberg, Germany). Each virtual thin section is 10 μm thick.

**Data availability.** All the relevant data that support this study are available from the corresponding author upon request.

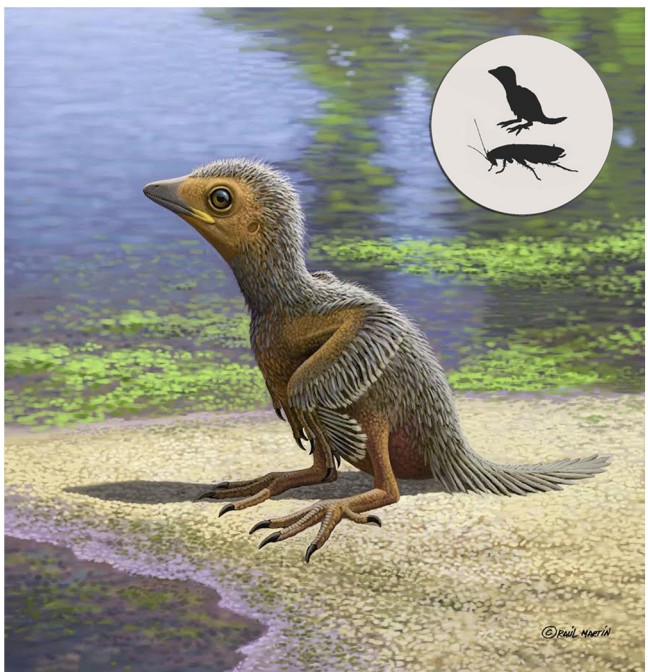

**Fig. 5** Hypothetical fleshed-out reconstruction of MPCM-LH-26189. The fact that MPCM-LH-26189 is so well preserved and conserves some soft-tissue associated chemistry (see Supplementary Note 1, Supplementary Figs 2-5, and Supplementary Table 2), while showing no feathers or chemical evidence for plumage suggest that the baby bird might have been largely featherless when it died. However, this cannot be confirmed, and so the individual was reconstructed with juvenile plumage hypothesized for perinate enantiornithines. The silhouettes inset are those of the juvenile bird and an unspecified sympatric cockroach to give a sense of scale. By Raúl Martín

### Table 1 Asynchronous ossification patterns in immature enantiornithine specimens

| | α | β | | γ | | δ | |
|---|---|---|---|---|---|---|---|
| | UFRJ-DG 031 Av[19] | PIN 4492/3[24] | GMV-2158[3] | MPCM-LH-26189 | GMV-2156/NIGP-130723[3] | GMV-2159[3] | STM 34-1[9] |
| Coracoid | 7.7 | 6.0 | 7.0 | 5.4 | 7.5 | 9.7 | — |
| Humerus | 14.0 | 13.0 | 15.6 | 10.6 | 15.4 | 20.6 | 20.4 |
| Ulna | 13.3 | 15.0 | 15.6 | 9.9 | 15.6 | 20.9 | 21.3 |
| Radius | 12.5 | 14.0 | 14.7 | 8.7 | 14.9 | 19.5 | 19.7 |
| Pubis | — | — | 9.5 | 6.5 | — | 12.5 | 15.3 |
| Ischium | — | — | 6.2 | — | — | — | — |
| Femur | 12.8 | — | 14.3 | 10.8 | 14.5 | 17.2 | 19.8 |
| Tibia | 12.0 | — | 18.0 | — | 16.8 | 20.8 | 25.4 |
| Free caudal vertebrae | 8 | ? | 8+ | 10+ | 7 | 7/8 | 7 |

Degree of ossification of the sternum: α → no sternal ossification; β → ossified xiphial region only; γ → ossified xiphial region and trabeculae; δ → xiphial region, trabeculae, and cranial disc. All the specimens are in the ontogenetic stage 0 (sensu ref. [25]). Note the lack of correlation between the ossification of the sternum, the size of the specimens, and the number of free caudal vertebrae
GMV, National Geological Museum of China, Beijing, China; MPCM, Museo de Paleontología de Castilla-La Mancha, Cuenca, Spain; NIGP, Nanjing Institute of Geology and Palaeontology, Nanjing, China; PIN, Borissiak Palaeontological Institute of the Russian Academy of Sciences, Moscow, Russia; STM, Tianyu Museum of Nature, Shandong, China; UFRJ-DG, Department of Geology of the Universidade Federal do Rio de Janeiro, Rio de Janeiro, Brazil

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

## Acknowledgements

We are grateful to the following persons, who have supplied practical assistance throughout the course of this project: S. Abramowicz (Natural History Museum of Los Angeles County, Los Angeles), R. Alonso-Mori (SLAC National Accelerator Laboratory, Menlo Park), U. Bergmann (SLAC National Accelerator Laboratory, Menlo Park), V. M. Egerton (College of Charleston, Charleston), A. García-Tabernero (Museo Nacional de Ciencias Naturales-CSIC, Madrid), T. Hayden (Natural History Museum of Los Angeles County, Los Angeles), R. López-Antoñanzas (Université de Montpellier, Montpellier), P. Peláez-Campomanes (Museo Nacional de Ciencias Naturales-CSIC, Madrid), A. Serrano-Martínez (Universidad Nacional de Educación a Distancia, Madrid), D. Sokaras (SLAC National Accelerator Laboratory, Menlo Park), and P. Tafforeau (European Synchrotron Radiation Facility, Grenoble). E. E. Maxwell (Staatliches Museum für Naturkunde, Stuttgart) and J.M. Starck (Ludwig-Maximilians-Universität München, Munich) kindly provided enlightenment on the timing of ossification in birds. J. Madero-Jarabo (Museo de las Ciencias de Castilla-La Mancha, Cuenca) allowed and facilitated our work on MPCM-LH-26189. This project was funded by the European Union within the framework of a Marie Curie individual fellowship (PIEF-GA-2013-624969 to F.K.) and the UK Natural Environment Research Council (NE/J023426/1 to R. A.W.). Additional support was provided by the European Synchrotron Radiation Facility through beam time allocated at the beamline ID19 and BM05 (ES-154 to F.K.). F. K. is an ARAID researcher.

## Author contributions

F.K., L.M.C., P.L.M. and J.L.S. designed the project. F.K and L.M.C. wrote the manuscript. F.K., L.M.C., S.S., R.J.G., N.P.E., R.A.W., W.I.S. and P.L.M. performed the analyses. S.S., R.J.G., N.P.E., R.A.W., W.I.S., P.L.M., F.O., F.J.S., J.M.-L. and J.L.S. commented on the manuscript. E.C. and F.E. took part in the fieldwork and/or curation of the specimen.
