## [Peer Review File · Nature Communications]

Reviewers' comments:

Reviewer #1 (Remarks to the Author):

The manuscript is clearly written. The fossil is exceptional for its size, ontogenetic stage, and preservation, bringing important information on the developmental evolution of birds. However, the description of the specimen is somewhat clouded by the authors' emphasis on the "asynchrony" between the development of the sternum and the pygostyle in different species of enantiornithes. This statement needs some clarifications:

1. Lines 127-129: "We suggest that this unusually high number of free caudal vertebrae is due to the immature stage of the specimen". The number of free caudal vertebrae varies from 4 to 10 in extant birds (Raikow, 1985). Most enantiornithes have 8. The authors infer that juveniles enantiornithes with different number of free caudal vertebrae are in different ontogenetic stages (127-132). How could it be discarded that the number of free vertebrae reflects interspecific variation?

2. Lines 129-130: "the caudal-most vertebrae were likely on their way to becoming fused with the pygostyle when the individual died". In extant birds, the pygostyle is patterned still during somitogenesis (Rashid et al., 2014; Tenin et al., 2010), and it is not composed by fully independent cartilages. Ossification centers in the pygostyle fuse very early, not as individual vertebrae articulated to each other (as observed in the specimen). Is there any evidence that in Enantiornithes the pygostyle develops by the late fusion of fully independent vertebrae?

3. Lines 130-132: "In early birds, like in extant taxa, there are a higher number of free caudal vertebrae in the juvenile stages of any given species, which is concomitantly reflected in the length of the pygostyle." Could you please provide some references for this statement? Is it possible that the smaller number of free vertebrae in older animals is due to the fusion of anterior caudal vertebrae to the synsacrum? (Notice that while there are two extra free caudal vertebrae, the specimen lacks two sacral vertebrae: "The sacrum appears to be composed of 5 to 6 vertebrae; fewer than the eight typical of the synsacrum of adult enantiornithines but similar to the six reported for other enantiornithine juveniles" SI lines 26-28).

4. Lines 152-154: "These observations are consistent with independence between the ossification sequences of the sternum and the tail in early juveniles Enantiornithes." In extant birds, the beginning of the ossification of the sternum varies from late embryogenesis to the first weeks after hatching (cf lines 134-135), even in closely related species, such as quails and chickens, for example. This can be inferred to be true also for enantiornithes from previously known fossils (cf lines 138-154). The chondrogenesis and ossification of the avian sternum depends on the action of pectoral and abdominal muscle, consequently, on muscular development and locomotor capabilities (cf lines 167-170). Is it appropriate to use such a phylogenetic and functionally plastic trait to infer variations on skeletal developmental sequences, instead of simply assuming that the ossification of the

sternum is highly variable and independent of the remaining skeleton?

I greatly enjoyed reading the anatomical description in the Supplementary Information. I suggest to move much of it to the "Results" section in the main text, especially those parts describing ontogenetically relevant traits, such as the number of vertebrae in the sacrum. It would contribute to the understating of some points, such as the importance of the number of free vertebrae in the tail, discussed in the "Discussion" section, but described only in the SI.

Likewise, some content in the M&M section called "Histological analysis" would fit better in the "Results" section (lines 212-238). I believe this would result in a more comprehensive description of an extraordinary fossil with important informations on the evolution of avian developmental modes, instead of aiming to highlight an unclear developmental trait.

Minor comments:

The first paragraph in the Introduction section would benefit from more references.

78-79 "the cerebral anatomy of MPCM-LH-26189 falls in between that of the basal avian Archaeopteryx and the putative early ornithurine Cerebavis". It would be nice to succinctly explain what this means. For instance, is the relative size of the telencephalon intermediary?

82-83 "Including the presence of two synapomorphies for this clade: the distal end of metacarpal III extending beyond the end of metacarpal II". It would be nice to number the MC in the figure.

The word "early" is used in the manuscript to refer to both phylogenetically basal ("early birds"), and ontogenetically young ("early juvenile enantiornithines"). It generates some ambiguities.

Reviewer #2 (Remarks to the Author):

I enjoyed reading this manuscript. The authors reported a perinate enantiornithine bird, and discussed the ossification sequence of the sternum and vertebrate column, showing that these body regions exhibit asynchronous ossification. Despite the wealthy materials of enantiornithines discovered recently, specimens recording such early ontogenetic stages are rare, severely limiting our understanding about the ontogeny of this important Mesozoic avian clade. Therefore, this important discovery provided valuable information about this issue, and is very suitable to be published in Nature Communications. I appreciate the authors that use advanced techniques such as synchrotron and element mapping to extract details that are not available using traditional methods. It is a very nice piece of work, and the conclusion is solid, and definitely benefits the whole community about the evolution of birds. But I do have some suggestions for minor revision, outlined below.

Page 3, line 67: "remarkable fossil", remarkable is too subjective, and may consider remove.

page 3, line 69: ".was facilitated by the fact that cranial sutural obliteration is rare among Enantiornithes": instead, the authors may say something like "fact that the cranial elements such as the frontals and parietals are not fused with one another in most enantiornithines"

page 4, line 81: "The gross skeletal morphology of MPCM-LH-26189 is consistent with that of other", instead, the authors may consider to say something that MPCM-LH-26189 can be safely referred to the Enantiornithes on basis of preserving two synapomorphies of this clade.

page 4, line 83 "metacarpal III extending beyond the end of metacarpal II". I would suggest use alular and major metacarpals in consistence with other ornithologists.

Page 4. line 98, "Fig 4". Actually it is Fig 3. Similar mistakes are also encountered in other paragraphs. Also, the citation of the panel in figure 3 (the histological one) is incorrect in many places and also Figure 3 caption.

Page 5, line 110. "plate-like sternum of other non-ornithuromorp birds", "non-ornithothoracine birds" is more appropriate here.

Page 125. line 125-127. I am concerned about the count of the free caudals. I agree that it appears more free caudals present in this bird. Is it possible that some of the cranial most ones belong to the sacral region, is there any morphologies of the vertebrae supporting their free caudal assignment? It would be more informative to draw a line in Figure 2, showing the boundary between the sacral region and the free caduals.

Page 6 line 145. ". tail ossification than GMV-2156". isn't "AS GMV-2156"?

Page 8. Histological analysis. Forgive if I was wrong. In figure 4b, the labeled "medullary cavity" is strange to me. The black dots/holes are the osteocyte lacunae? If so, does it indicate that the medullary cavity is not free of bone elements as in other more mature enantiornithines?

Last. Is there any living vertebrates that show synchronous ossification in sternum and vertebral column? If so, what is underlying biological significance for appearance of asynchronous ossification present in birds? If not, it may reflect the fact that asynchronous ossification is widely distributed in animals but not exclusively to birds. I hope the authors can elaborate this issue, and that would certainly increase significance of this work.

I'd be happy to reveal my identity to the authors. Hope the authors find these comments helpful.

Min Wang

Reviewers' comments:

Reviewer #1:

The manuscript is clearly written. The fossil is exceptional for its size, ontogenetic stage, and preservation, bringing important information on the developmental evolution of birds.

However, the description of the specimen is somewhat clouded by the authors' emphasis on the "asynchrony" between the development of the sternum and the pygostyle in different species of enantiornithes. This statement needs some clarifications:

1. Lines 127-129: "We suggest that this unusually high number of free caudal vertebrae is due to the immature stage of the specimen". The number of free caudal vertebrae varies from 4 to 10 in extant birds (Raikow, 1985). Most enantiornithes have 8. The authors infer that juveniles enantiornithes with different number of free caudal vertebrae are in different ontogenetic stages (127-132). How could it be discarded that the number of free vertebrae reflects interspecific variation?

Reply:

This is an interesting question. Actually, in all but one of the juvenile specimens studied in our manuscript the number of free caudal vertebrae is about 7/8. This is in line with what we observe in adult Enantiornithes, so we assume that in these specimens the number of free caudal vertebrae seen at the adult stage was already reached. The only exception is the specimen from Las Hoyas. In this specimen, there were at least 10 free caudal vertebrae. To our knowledge, this is well above what is observed in any adult Enantiornithes. So, even though there was certainly some interspecific variation in the number of free caudal vertebrae in Enantiornithes (evidently lower than that seen in modern birds), the most reasonable interpretation here is that the unusually high number of free caudal vertebrae in the very young Enantiornithes from Las Hoyas is a result of its immaturity.

2. Lines 129-130: "the caudal-most vertebrae were likely on their way to becoming fused with the pygostyle when the individual died". In extant birds, the pygostyle is patterned still during somitogenesis (Rashid et al., 2014; Tenin et al., 2010), and it is not composed by fully independent cartilages. Ossification centers in the pygostyle fuse very early, not as individual vertebrae articulated to each other (as observed in the specimen). Is there any evidence that in Enantiornithes the pygostyle develops by the late fusion of fully independent vertebrae?

Reply:

This is again a very pertinent question. Unfortunately, we do not have a wealth of data regarding the mechanism of pygostyle formation in Enantiornithes. One of the most relevant specimens is the one from Las Hoyas that we report herein: this is one of the only specimens known in which the ossification of the tail is significantly different from what would be expected from an adult. In most other juvenile Enantiornithes specimens found to date (which is relatively limited), the state of ossification of the tail is basically that of an adult.

3. Lines 130-132: "In early birds, like in extant taxa, there are a higher number of free caudal vertebrae in the juvenile stages of any given species, which is concomitantly reflected in the length of the pygostyle." Could you please provide some references for this statement? Is it possible that the smaller number of free vertebrae in older animals is due to the fusion of anterior caudal vertebrae to the synsacrum? (Notice that while there are two extra free caudal vertebrae, the specimen lacks two sacral vertebrae: "The sacrum appears to be composed of 5 to 6 vertebrae; fewer than the eight typical of

the synsacrum of adult enantiornithines but similar to the six reported for other enantiornithine juveniles” SI lines 26-28).

Reply:

We have removed this sentence, as a number of free caudal vertebrae in hatchling birds do indeed fuse eventually with both the synsacrum and pygostyle. Rather we have modified the passage to make it clear that the two or more extra free caudals were not necessarily destined to fuse with the pygostyle. This does not change our argument, but this clarification was necessary. We are grateful to the reviewer for this insight.

4. Lines 152-154: “These observations are consistent with independence between the ossification sequences of the sternum and the tail in early juveniles Enantiornithes.” In extant birds, the beginning of the ossification of the sternum varies from late embryogenesis to the first weeks after hatching (cf lines 134-135), even in closely related species, such as quails and chickens, for example. This can be inferred to be true also for enantiornithes from previously known fossils (cf lines 138-154). The chondrogenesis and ossification of the avian sternum depends on the action of pectoral and abdominal muscle, consequently, on muscular development and locomotor capabilities (cf lines 167-170). Is it appropriate to use such a phylogenetic and functionally plastic trait to infer variations on skeletal developmental sequences, instead of simply assuming that the ossification of the sternum is highly variable and independent of the remaining skeleton?

Reply:

The reviewer is correct again, and we are grateful for this point. The timing of sternum ossification varies between species in extant birds, independently of their phylogenetic relationships, but rather in relation to functional demands or other factors. In our paper, we do not give sternum ossification greater weight, but do stress that such a modern avian character was already present in a very primitive group of birds (because of the extreme rarity of fossil hatchlings of primitive birds, this has never been documented before, and we believe it is thus important to highlight). We have modified this passage to make it clearer.

I greatly enjoyed reading the anatomical description in the Supplementary Information. I suggest to move much of it to the “Results” section in the main text, especially those parts describing ontogenetically relevant traits, such as the number of vertebrae in the sacrum. It would contribute to the understating of some points, such as the importance of the number of free vertebrae in the tail, discussed in the “Discussion” section, but described only in the SI.

Reply:

We have moved all of the “additional anatomical description” of the “Supplementary Information” to the body of the text in the new version: thanks.

Likewise, some content in the M&M section called “Histological analysis” would fit better in the “Results” section (lines 212-238). I believe this would result in a more comprehensive description of an extraordinary fossil with important informations on the evolution of avian developmental modes, instead of aiming to highlight an unclear developmental trait.

Reply:

We agree and have moved most of “Histological analysis” of the “Methods” section to the body of the text in the new version.

Minor comments:

The first paragraph in the Introduction section would benefit from more references.

Reply:

The introduction has been improved in the new version and two references were added.

78-79 “the cerebral anatomy of MPCM-LH-26189 falls in between that of the basal avian Archaeopteryx and the putative early ornithurine Cerebavis”. It would be nice to succinctly explain what this means. For instance, is the relative size of the telencephalon intermediary?

Reply:

This is done in the new version.

82-83 “Including the presence of two synapomorphies for this clade: the distal end of metacarpal III extending beyond the end of metacarpal II”. It would be nice to number the MC in the figure.

Reply:

We have indicated that what we label in Fig. 1 is the MC III. The other MC may be the MC II as we initially thought, but are no longer completely certain of this identification, we left it unlabelled and changed our text accordingly in the new version. This has no impact on the identification of our specimen as an Enantiornithe as attested by the lateral excavation of the furcular rami.

The word “early” is used in the manuscript to refer to both phylogenetically basal (“early birds”), and ontogenetically young (“early juvenile enantiornithines”). It generates some ambiguities.

Reply:

This is corrected in the new version.

Reviewer #2:

I enjoyed reading this manuscript. The authors reported a perinate enantiornithine bird, and discussed the ossification sequence of the sternum and vertebrate column, showing that these body regions exhibit asynchronous ossification. Despite the wealthy materials of enantiornithines discovered recently, specimens recording such early ontogenetic stages are rare, severely limiting our understanding about the ontogeny of this important Mesozoic avian clade. Therefore, this important discovery provided valuable information about this issue, and is very suitable to be published in Nature Communications. I appreciate the authors that use advanced techniques such as synchrotron and element mapping to extract details that are not available using traditional methods. It is a very nice piece of work, and the conclusion is solid, and definitely benefits the whole community about the evolution of birds. But I do have some suggestions for minor revision, outlined below.

Page 3, line 67: “remarkable fossil”, remarkable is too subjective, and may consider remove.

Reply:

This is removed in the new version.

page 3, line 69: “..was facilitated by the fact that cranial sutural obliteration is rare among Enantiornithes”: instead, the authors may say something like “fact that the

cranial elements such as the frontals and parietals are not fused with one another in most enantiornithines”

Reply:

This is done in the new version.

page 4, line 81: “The gross skeletal morphology of MPCM-LH-26189 is consistent with that of other”, instead, the authors may consider to say something that MPCM-LH-26189 can be safely referred to the Enantiornithes on basis of preserving two synapomorphies of this clade.

Reply:

This is done in the new version.

page 4, line 83 “metacarpal III extending beyond the end of metacarpal II”. I would suggest use alular and major metacarpals in consistence with other ornithologists.

Reply:

This is done in the new version.

Page 4. line 98, “Fig 4”. Actually it is Fig 3. Similar mistakes are also encountered in other paragraphs. Also, the citation of the panel in figure 3 (the histological one) is incorrect in many places and also Figure 3 caption.

Reply:

All of this is corrected in the new version.

Page 5, line 110. “plate-like sternum of other non-ornithuromorp birds”, “non-ornithothoracine brids” is more appropriate here.

Reply:

This is done in the new version.

Page 125. line 125-127. I am concerned about the count of the free caudals. I agree that it appears more free caudals present in this bird. Is it possible that some of the cranial most ones belong to the sacral region, is there any morphologies of the vertebrae supporting their free caudal assignment? It would be more informative to draw a line in Figure 2, showing the boundary between the sacral region and the free caudals.

Reply:

Yes, indeed. The first free caudal is likely to have been incorporated into the synsacrum during ontogeny. We have now indicated in Fig. 1 the exact extent of each vertebral series, including the sacral zone, with brackets - thanks for this suggestion.

Page 6 line 145. “.. tail ossification than GMV-2156”. isn’t “AS GMV-2156”?

Reply:

Yes. This is corrected in the new version.

Page 8. Histological analysis. Forgive if I was wrong. In figure 4b, the labeled “medullary cavity” is strange to me. The black dots/holes are the osteocyte lacunae? If so, does it indicate that the medullary cavity is not free of bone elements as in other more mature enantiornithines?

Reply:

In Fig. 3 (it is actually not Fig. 4 – we have updated that), bone and matrix are of about the same shade of grey, due to similar densities and thus similar X-ray attenuation coefficients. During the fossilisation process, the empty medullary cavity was filled in with rock matrix,

which therefore appears close in grey value to bone. The inner surface of the bone is, however, distinct as a result of the phase contrast techniques employed, which are particularly well suited to resolving interfaces. We can, therefore, see that the medullary cavity is fully devoid of bone. The black dots in the rock matrix do not represent osteocyte lacunae, but rather empty microcavities, or microcavities infilled with low density inclusions within the sediment. In order to clarify this point, the caption of Fig. 3 has been modified.

Last. Is there any living vertebrates that show synchronous ossification in sternum and vertebral column? If so, what is underlying biological significance for appearance of asynchronous ossification present in birds? If not, it may reflect the fact that asynchronous ossification is widely distributed in animals but not exclusively to birds. I hope the authors can elaborate this issue, and that would certainly increase significance of this work.

Reply:

How ossification sequences are conserved in amniotes is a highly active question in evo-devo currently. The vertebrate sternum exists only in tetrapods, and it generally remains cartilaginous in amphibians and reptiles. In birds, clarity of the patterns in sternum development is difficult because different parts of the sternum ossify at different time. In mammals, the relative timing of ossifications varies significantly between taxa (Hautier et al., 2011: *Evolution & Development* 13: 460–476). As such, we believe it is too early to make meaningful comparisons between high-level taxonomic groups, and draw sound general conclusions from these. We believe, however, that our work is one step towards that goal.

REVIEWERS' COMMENTS:

Reviewer #1 (Remarks to the Author):

The manuscript has been significantly improved. The description of the specimen in the results section is more complete; methods section is actually restricted to methods; most important, the discussion about the development and evolution of pygostyle is now flawless and well-grounded.

I recommend this paper for publication without further revisions.

Reviewer #2 (Remarks to the Author):

I thank the authors for clarifying all my questions.

Just one sentence needs to be revised; line 100: The thoracic centra appear excavated by a broad but relatively shallow fossa.

You mean the lateral facet of the centrum is excavated, right? if so, make it clear.

I think the manuscript is ready for publication.

Reviewers' comments:

Reviewer #1 (Remarks to the Author):

The manuscript has been significantly improved. The description of the specimen in the results section is more complete; methods section is actually restricted to methods; most important, the discussion about the development and evolution of pygostyle is now flawless and well-grounded.

I recommend this paper for publication without further revisions.

Reply:

We are grateful for this comment, and the reviewer's previous suggestions.

Reviewer #2 (Remarks to the Author):

I thank the authors for clarifying all my questions.

Just one sentence needs to be revised; line 100: The thoracic centra appear excavated by a broad but relatively shallow fossa.

You mean the lateral facet of the centrum is excavated, right? if so, make it clear.

I think the manuscript is ready for publication.

Reply:

Yes, the lateral excavation of the centra is a common feature among enantiornithines (and some other early birds too). It is a recess (of variable shape depending on the taxa and the position of the vertebra) that excavates the *sides* of the centrum. We have modified this sentence (by adding "laterally") to make it clear.